# Kawasaki Shock Syndrome with Initial Presentation as Neck lymphadenitis: A Case Report

**DOI:** 10.3390/children9010056

**Published:** 2022-01-03

**Authors:** Yi-Ting Cheng, Yu-Shin Lee, Jainn-Jim Lin, Hung-Tao Chung, Yhu-Chering Huang, Kuan-Wen Su

**Affiliations:** 1Department of Pediatrics, Chang Gung Memorial Hospital, Taoyuan City 333423, Taiwan; b9902093@cgmh.org.tw (Y.-T.C.); b9202021@cgmh.org.tw (Y.-S.L.); lin0227@cgmh.org.tw (J.-J.L.); hungtao@cgmh.org.tw (H.-T.C.); ychuang@cgmh.org.tw (Y.-C.H.); 2Department of Pediatrics, Keelung Chang Gung Memorial Hospital, Chang Gung University, Taoyuan City 333423, Taiwan

**Keywords:** cardiogenic shock, cervical lymphadenitis, Kawasaki disease, mucocutaneous lymph node syndrome

## Abstract

Kawasaki disease (KD) is an acute systemic vasculitis of unknown cause that mainly affects infants and children and can result in coronary artery complications if left untreated. A small subset of KD patients with fever and cervical lymphadenitis has been reported as node-first-presenting KD (NFKD). This type of KD commonly affects the older pediatric population with a more intense inflammatory process. Considering its unusual initial presentation, a delay in diagnosis and treatment increases the risk of coronary artery complications. Herein, we report the case of a 9-year-old female with fever and neck mass that rapidly deteriorated to shock status. A diagnosis of KD was made after the signs and symptoms fulfilled the principal diagnostic criteria. The patient’s heart failure and blood pressure improved dramatically after a single dose of intravenous immunoglobulin. This case reminds us that NFKD could be the initial manifestation of KDSS, which is a potentially fatal condition. We review the literature to identify the overlapping characteristics of NFKD and KDSS, and to highlight the importance of early recognition of atypical KD regardless of age. We conclude that unusually high C-reactive protein, neutrophilia, and thrombocytopenia serve as supplemental laboratory indicators for early identification of KDSS in patients with NFKD.

## 1. Introduction

Kawasaki disease (KD) is an acute systemic vasculitis that mainly affects medium-sized vessels [1]. It occurs predominantly in children less than 5 years of age and results in coronary artery abnormalities in up to 25% of untreated children [2]. The global prevalence of KD among children is the highest in Japan (218/100,000) and the lowest (4.7/100,000) in European descent. Taiwan has the third-highest incidence of KD in the world, following Japan and Korea [3]. Diagnosis is based on nonspecific clinical signs, including fever for at least 5 days, bilateral bulbar conjunctival injection, changes in lips and oral cavities, polymorphous rash, changes in extremities, and cervical lymphadenopathy [4]. In patients who lack sufficient clinical signs of the disease to fulfill the classic criteria, “incomplete (atypical) KD” is suspected if only two to three clinical criteria are observed on the 5th day of fever. Further supplemental laboratory criteria (low albumin, anemia, elevated ALT, thrombocytosis, leukocytosis, pyuria) should be taken to make a diagnosis and facilitate prompt treatment [5]. Even for experienced clinicians, the atypical presentation of KD can be challenging for a prompt diagnosis. Previous studies showed that node-first-presenting KD (NFKD) is characterized by fever and cervical lymphadenopathy before other clinical features appear [6]. Some patients with KD may also present with hypotension or shock, known as KD shock syndrome (KDSS) [7]. Diagnosing NFKD and KDSS patients is more difficult and usually delayed, leading to more coronary artery complications. Here, we report a rare case of KD at an unusual older age, manifesting as both NFKD and KDSS. A literature review was performed to determine the clinical characteristics of NFKD and KDSS, and to increase the awareness of early identification of KDSS in NFKD cases during daily clinical practice.

## 2. Case Presentation

A previously healthy 9-year-6-month-old Taiwanese female was admitted to the hospital with a 3-day history of progressive left neck pain and fever. The patient had neck pain and a palpable neck mass one day before fever developed. She had no skin rash before admission. The patient denied having had contact with any person with severe acute respiratory syndrome coronavirus 2 (SARS-CoV-2) infection within a few months preceding admission. Upon admission (on the third day of fever), the patient’s body temperature was 39.4 °C, heart rate was 124 beats/min, blood pressure was 106/53 mmHg, and respiratory rate was 17 breaths/min. Physical examination revealed bilaterally injected tonsils with exudate. A firm, tender mass, approximately 4 cm in diameter, was palpable on the left side of the anterior cervical region. Erythematous macular skin rashes scattered over the patient’s bilateral lower legs were observed on the day of admission. Laboratory examination showed a white blood cell count within the normal limit (6600/uL), a normal platelet count (208,000/uL), but an elevated C-reactive protein (CRP) level (29.13 mg/L). Urine sediment contained 30 white cells per micro-liter.

We initially diagnosed the patient with acute exudative tonsillitis and cervical lymphadenitis, and the patient was then treated with intravenous antibiotics (amoxicillin/clavulanate potassium). On the second hospital day, the patient developed vomiting, watery diarrhea, and simultaneous spread of skin rashes over the bilateral arms and the abdomen. On the third hospital day, the patient suddenly presented with poor activity and hypotension (blood pressure dropped to 86/43 mmHg). Fluid resuscitation and inotropic agents were provided. At that point, laboratory examination showed a remarkable elevation in CRP (202.38 mg/L), leukocytosis (white blood cell count 13,900/µL) with 86.5% neutrophils, thrombocytopenia (platelet count 108 k/uL), and hyperlactatemia (lactate 2.45 mmol/L). For septic shock, the antibiotics were substituted with vancomycin and piperacillin/tazobactam. To identify the source of infection, computed tomography of the head and neck was performed, disclosing multiple enlarged cervical lymph nodes along the left jugular chain. Chest radiography showed obvious cardiomegaly compared to the previous radiograph that was taken upon admission. The patient’s N-terminal pro-brain natriuretic peptide level (6559 pmol/L) and troponin-I level (0.829 µg/L) were both elevated; therefore, KDSS was suspected. Echocardiography revealed mildly reduced left ventricular contraction (ejection fraction, 57%) and pericardial effusion; however, no coronary artery dilatation was detected. The results for the real-time reverse transcription-polymerase chain reaction for severe acute respiratory syndrome coronavirus 2 and other serological tests for Epstein–Barr virus, Mycoplasma, and group A Streptococcus were all negative.

On the sixth hospital day, bulbar conjunctival injection, fissured lips, and erythema with edematous changes over the bilateral palms and feet developed. Subsequently, the patient was diagnosed with KD. Hence, intravenous immunoglobulin (IVIG) (1 g/kg/day for 2 days) and aspirin (30 mg/kg/day for 3 days) were administered, which promptly relieved the patient’s fever and KD symptoms. Hypotension subsided gradually, and inotropic agents were discontinued on the tenth hospital day. The patient’s clinical manifestations, laboratory data, and management are summarized in Figure 1. The electrocardiography and echocardiogram obtained before discharge were normal. However, the Z-score remarkably decreased by 1 to 1.7 units on day 14, which was defined as resolved dilation according to the American Heart Association (AHA) criteria [8]. The patient was discharged on day 16 and was prescribed with low-dose aspirin (3 mg/kg/day). Echocardiography conducted on day 48 did not reveal any coronary dilation, and low-dose aspirin was discontinued 2 months later. Subsequent echocardiography at 6 months post-discharge was normal and without any coronary artery lesion. The echocardiographic features are summarized in Table 1.

## 3. Discussion

Among the principal diagnostic criteria for KD, cervical lymphadenopathy is the least common presentation [6]. Among typical KD patients, 8–21% have been classified as NFKD and have presented with only cervical lymphadenopathy and fever in their initial visit [6,9,10,11,12]. This group of patients has often been misdiagnosed with suppurative cervical lymphadenitis, deep neck infection, or cellulitis [13]. Based on previous studies, patients with NFKD may require an additional 0.5–3 days to make a definitive diagnosis of KD [6,9,11,12]. Kanegaye et al. [6] compared 57 patients with NFKD to 78 patients with a final diagnosis of bacterial cervical lymphadenitis (BCL). They found that patients with NFKD had significantly higher absolute neutrophil counts, CRP, and alanine transaminase levels than those of the BCL group. In imaging studies, including computed tomography or ultrasound, patients with NFKD more commonly had a cluster of lymph nodes of different sizes, whereas patients with BCL had mostly phlegmon or abscesses.

April et al. [14] found that KD patients with cervical lymphadenopathy were older and had a stronger inflammatory response than those without lymphadenopathy. Similarly, patients with NFKD were significantly older (3.9–4.9 year-old versus 1–2.4 year-old) and demonstrated a higher CRP level (76–133 mg/L) and neutrophil percentage (76–81%) or absolute neutrophil count, compared to other KD patients. [10,11,12]. These findings support NFKD as a more severe form of KD with a more intense inflammatory process.

The association between NFKD and KDSS remains unclear, but a case of KDSS with retropharyngeal edema has been reported previously [15]. As in this case report, the patient manifested hypotension and shock after admission, meeting the definition for KDSS. In Western countries, the prevalence of KDSS has been reported as up to 5% of patients with KD [16], while a large-scale population-based epidemiological report in Taiwan disclosed a relatively lower prevalence of only 1.45% [1]. While most KD cases are diagnosed in children younger than 5 years of age, previous studies recognize 10 years of age and older as a risk factor for KDSS [2]. In Taiwan, a significantly higher incidence of KDSS was noted in KD patients aged 5 years and older, with a peak age at 8–9 years [17]. Patients with KDSS have significantly higher rates of gastrointestinal symptoms, respiratory symptoms, pleural effusion, and compromised renal function than patients with typical KD [2]. Since KDSS patients initially present with less frequent manifestations rather than typical Kawasaki criteria, it is a challenge for clinicians to make a prompt KDSS diagnosis and provide early IVIG treatment [16].

The mechanism of KDSS is still under investigation. Patients with KDSS are reported to have higher CRP levels, lower hemoglobin levels, hyponatremia, and hypoalbuminemia, as well as elevated markers of myocardial damage [16]. Intensive vasculitis with capillary leak and myocardial dysfunction caused by the release of proinflammatory cytokines is the probable mechanism [16]. Furthermore, Kuo et al. [1] reported that 26 KD patients in Taiwan who required pediatric intensive care tended to have a lower platelet count upon admission and reached a nadir of thrombocytopenia with platelet count <150 × 10^9^ L after admission.

During the recent SARS-CoV-2 pandemic, a novel syndrome called “multisystem inflammatory syndrome in children (MIS-C)” was firstly identified in April 2020 in the United Kingdom. MIS-C has several characteristics similar to KDSS, including a higher rate to present with gastrointestinal symptoms, cardiac impairment or shock, hyponatremia, and a higher rate of IVIG resistance [18]. The overlapping of the clinical manifestations makes us reconsider the mechanism behinds KD and MIS-C. Firstly, infectious etiology, such as some common respiratory viruses, is reported to be one of the predisposing factors of KD [19]. MIS-C cases appear 1 month after the SARS-CoV-2 peak in the population, having a low proportion of positive results by real-time polymerase chain reaction (RT-PCR), while having a high proportion of positive results by serological testing [18]. Interestingly, a recent research study in a single institution of Taiwan observed a 40% decline in KD incidence during the SARS-CoV-2 pandemic in Taiwan. This may be contributed to aggressive and strict COVID-19 prevention measures, including hand-washing, mask-wearing, and social-distancing practices in Taiwan since early 2020 [3]. The evidence supports that this inflammatory syndrome is caused by post-infectious immune dysregulation, rather than direct infection. Secondarily, MIS-C may share similar features of the inflammatory process as documented in KD. A virus might act as the immune trigger, facilitate the formation of immune complexes, and activate the development of T-cell immune responses [18]. Finally, genetic predisposition might play a role for both KD and MIS-C because KD is most common in Japanese children, while MIS-C is more prevalent in Hispanic and black children [20]. In this case, several atypical features remind us to take MIS-C into the list of differential diagnosis, such as older age, gastrointestinal involvement, rapidly progressing shock, hematological findings including neutrophilia, lymphopenia, and thrombocytopenia. The RT-PCR for SARS-CoV-2 infection in this case was negative. Besides, Taiwan remained a low incidence area during the SARS-CoV-2 pandemic. Prior to the patient’s illness onset, which was in May 2020, there were only 3 and 27 domestic SARS-CoV-2 cases in Taiwan in April and March, respectively. Hence, the patient with KDSS was favored rather than MIS-C.

Patients with KDSS are at higher risk of resistance to IVIG, coronary artery dilatation, and aneurysm [7]. Jun et al. found that the incidence of coronary artery lesions was more common in the NFKD group during the acute phase, while a lower incidence was found during convalesce stage. In the early stage of coronary artery lesion, neutrophils destruct the internal elastic lamina, followed by myofibroblast proliferation and formation of coronary artery aneurysm develops eventually. It is hypothesized that prompt treatment before the irreversible pathological change could prevent chronic coronary artery lesions. [9]. Most clinicians agree that delayed diagnosis is a risk factor for both IVIG resistance and coronary artery complications [10,11,12]. In this case, one dose of IVIG was administered on the sixth day after admission (the tenth day of illness). The serial measurements of coronary artery dimensions demonstrated a reduction in the Z-score ≥ 1 unit. This was compatible with the definition of coronary artery dilation by the 2017 AHA guideline [8]. Fortunately, this patient’s coronary artery dilation resolved within 4 weeks as the majority did. No additional IVIG or corticosteroid was required. Six months after discharge, the patient’s coronary artery was stable without dilatation.

## 4. Conclusions

In conclusion, this case emphasizes that NFKD can deteriorate into KDSS. Even though KD is more common in children less than 5 years of age, KD should be considered as a possible differential diagnosis in antibiotic-resistant cervical lymphadenitis or as a rapidly progressive shock syndrome, regardless of age. Early recognition of KD despite its unusual presentation prevents the risk of coronary artery lesions and IVIG resistance. Although NFKD is a less frequent presentation of KD, an unusually high CRP level with a high percentage of neutrophils and a decline in platelet count is a warning sign for the development of KDSS.

## Figures and Tables

**Figure 1 children-09-00056-f001:**
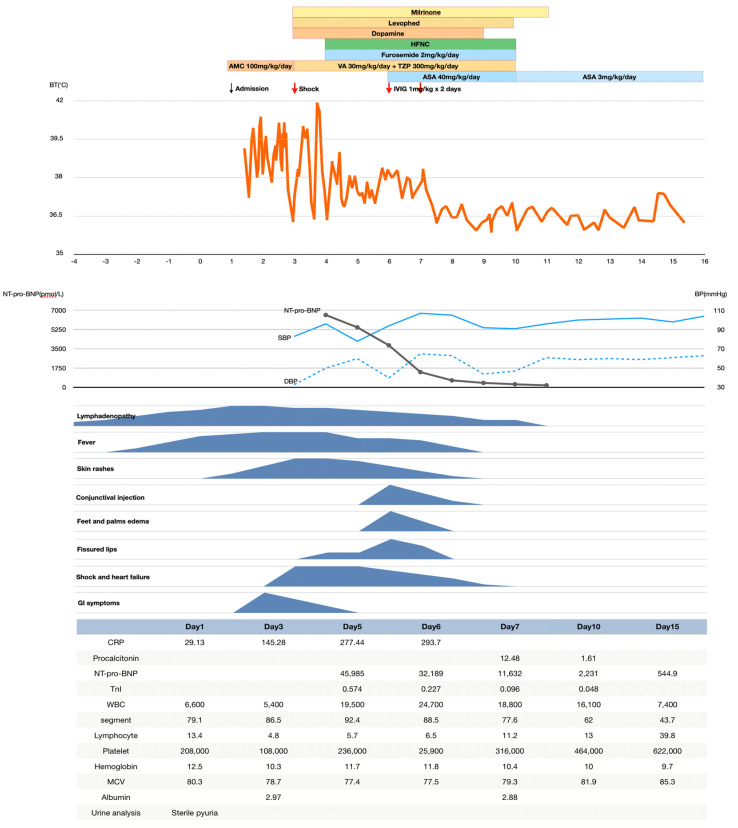
An overview of the patient’s clinical presentation, body temperature, blood pressure, laboratory data, and treatment. The patient had lymphadenopathy one day before fever developed, which is unusual for Kawasaki disease. The patient was treated as a case of acute exudative tonsillitis and cervical lymphadenitis. However, shock developed on the third hospital day even under intravenous antibiotics treatment. Thrombocytopenia was noticed on the third hospital day along with the shock. On the sixth hospital day, the patient fulfilled the diagnosis criteria of Kawasaki disease. Intravenous immunoglobulin and aspirin were administered, which promptly relieved the patient’s fever, hypotension, and KD symptoms. Fortunately, no coronary artery anomaly was revealed on day 48, and low-dose aspirin was discontinued 2 months later. (AMC: amoxicillin/clavulanate potassium; ASA: aspirin; BP: blood pressure; BT: blood temperature; CRP: C-reactive protein; DBP: diastolic blood pressure; HFNC: high-flow nasal cannula; IVIG: intravenous immunoglobulin; MCV: mean corpuscular volume; NT-proBNP: N-terminal pro-brain natriuretic peptide level; SBP: systolic blood pressure; TnI: troponin-I; TZP: piperacillin/tazobactam; VA: vancomycin; WBC: white blood cell).

**Table 1 children-09-00056-t001:** The serial echocardiographic features of the patient. (LMCA: left main coronary artery, LAD: left anterior descending artery, RCA: right coronary artery).

Z Score (Coronary Artery Size)	Day 6	Day 14	Day 48	6 Months Later
**LMCA**	+1.61 (3.4 mm)	+0.67 (2.51 mm)	+0.52 (2.46 mm)	−0.66 (2.37 mm)
**LAD**	+1.67 (2.95 mm)	+0.11 (2.46 mm)	+0.32 (2.55 mm)	−0.11 (2.37 mm)
**RCA**	+1.91 (3.25 mm)	60%	80%	74%
**Ejection fraction (%)**	57%	-	-	-
**Pericardial effusion**	+	-	-	-

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
