# Peer review of "Kawasaki Shock Syndrome with Initial Presentation as Neck lymphadenitis: A Case Report"

_children, 2022, doi:10.3390/children9010056_

Round 1

Reviewer 1 Report

The authors report a case of a nine-year-old with an initially unrecognized node first Kawasaki disease (NFKD) which evolved into a hemodynamically unstable state requiring fluids and vasoactive agents before being recognized as Kawasaki disease because of the evolution other findings.  The case is interesting and well described and would present diagnostic difficulty even at centers which treat Kawasaki disease regularly.

One of the important claims made by the authors is that this is the first reported case of NFKD with KDSS (discussion line 90), or at least the first reported case in Taiwan (abstract line 23). I relatively easily found the following case reporting from Taiwan e-published in 2012 and appearing in 2014:

Fang LC, et al. Unusual manifestations of Kawasaki disease with retropharyngeal edema and shock syndrome in a Taiwanese child. J Microbiol Immunol Infect. 2014 Apr;47(2):152-7. Epub 2012 Apr 12. PMID: 22503799.

Although the title makes no mention of NFKD, retropharyngeal edema is a known accompaniment to NFKD, and the abstract and report describe the presence of cervical adenopathy.  The claims for first reporting need to be revised, and the discussion should include mention of this published case.

I have a few other minor comments to add nuance to this report:

Abstract, line 16, implies causation ( increases the risk of coronary artery complications)  and should perhaps be stated as an association.

Line 22:  our other lab abnormalities worth mentioning as aids to early recognition?

Case: is it worth mentioning the patient’s menstrual  status and tampon use?

Line 45:   the timing of illness would be a bit more clear if it specified weather the presentation was on the third or fourth day of fever.

Line 52:  the description of rash at admission could raise the question of whether this was truly a node-first presentation unless the Report distinctly describes that the rash appeared later.

 Line 53: since thrombocytopenia is an important part of the case progression, it might be of interest to provide the platelet count on admission.

Line 55: the reporting of urinalysis seems a little unusual, at least buy North American conventions. “Sediment” implies a centrifuged specimen, usually reported in cells per high-powered field,  where asked him a cytometer accounts in cells per microliter implies an unspun specimen.

 Echocardiography (lines 71-2, 83, 87) provided reassurance against coronary abnormalities, but were Z-scores available for comparison overtime? Z scores, if they decreased over time, could suggest unrecognized coronary artery on the ability on the first study.

79-80:  readers might want to know the doses per kilogram of IVIG and aspirin, which are likely more important than the reported doses of amoxicillin-clavulanate)

 Lines 116 through 121:  perhaps a formatting issue, but unusual to have abbreviations appear at this location

 Lines 112 to 123: I’m sure the authors have recognized that April et all did not specifically describe adenopathy in NFKD ( rather they were describing the appearance of adenopathy among all patients with KD), but the less-careful reader might assume so based on how the sentences flow together.

Lines 137 to 138:  probably more accurate to use a term “findings”  or “criteria” rather than symptoms. Is there a reference to support this statement of presenting with only 1-2 findings?

Lines 150 to 151:  the authors might mention that the references are specifically to NFKD. Would it be more generalizable to use the term “IVIG resistance” rather than “repetitive I VIG,” because some centers might use a different agent rather than a second dose of IVIG?

 Line 157: it is unusual to make the first mention of “potentially fatal” nature of KDSS in the Conclusion ( might be more useful for to appear with reference in an earlier section)

 Lines 162 161: could this sentence be made more specific to NFKD and KDSS?

Reviewer 2 Report

The authors present a case report of a 9-year-old girl presenting with "node-first-presenting Kawasaki Disease" (NFKD), whereby neck lymphadenitis is the primary symptom. Patients with NFKD more commonly present with a severe disease course. In the case report, the patient went into shock, which dramatically improved after the admission of intravenous immunoglobulin. I thank the authors for sharing their report, which is an important addition to the limited literature on shock in KD patients. I have some comments to improve the manuscript:

  1. Introduction: please add a brief discussion on the global epidemiology of KD to better inform the reader.
  2. Lines 33-34: can the authors comment (and for the reader, specify) what atypical KD is and what providers need to pay particular attention to? This is particularly relevant considering the higher likelihood of shock in atypical KD patients, as briefly discussed on lines 137-139.
  3. While the patient denied having had contact with a COVID-19-positive patient, they may be unaware, whereas children are often asymptomatic. Was the patient tested (PCR and/or antibodies) to confirm?

Reviewer 3 Report

This is an interesting, descriptive case report of a patient with KDSS and NFKD.

A few comments:

Case Presentation:

Do you know if the patient had any kind of illness/infection in the month prior to KD onset?

The phrase "The patient denied having had contact with any coronavirus disease in 2019 within a few months..." is awkward. Would "The patient denied having had contact with any person with COVID-19 within a few months..." be better?

Discussion: Are you able to provide any information about how prevalent COVID-19 was in Taiwan in the month prior to the patient's illness onset? 

Round 2

Reviewer 1 Report

Thank you for making important revisions to the manuscript. I have only a few comments:

Because rash was present at admission,  mentioning that rash was absent at onset of fever and adenopathy (if true) would strengthen the node-first status.

Table 1 provides useful new information on coronary artery status. All Z-scores are <2, but all decreased by at least 1 unit, which may qualify as resolved dilitation according to the Z-score classification in AHA criteria (McCrindle, 2017, p e942). This change may warrant brief mention in the case and discussion, as decrease in artery diameter, even within the normal range, gives echo support to the diagnosis of KD and risk of coronary complications in KDSS and NFKD.

Line 275: NFKD is a "presentation" of KD and not a "criteria [sic]."

A few minor typographical errors made it into the revisions.
